META-RESEARCH

# COVID-19 research risks ignoring important host genes due to pre-established research patterns

**Abstract** It is known that research into human genes is heavily skewed towards genes that have been widely studied for decades, including many genes that were being studied before the productive phase of the Human Genome Project. This means that the genes most frequently investigated by the research community tend to be only marginally more important to human physiology and disease than a random selection of genes. Based on an analysis of 10,395 research publications about SARS-CoV-2 that mention at least one human gene, we report here that the COVID-19 literature up to mid-October 2020 follows a similar pattern. This means that a large number of host genes that have been implicated in SARS-CoV-2 infection by four genome-wide studies remain unstudied. While quantifying the consequences of this neglect is not possible, they could be significant.

**THOMAS STOEGER\* AND LUÍS A NUNES AMARAL\***

**\*For correspondence:** thomas. stoeger@northwestern.edu (TS); amaral@northwestern.edu (LANA)

**Competing interests:** The authors declare that no competing interests exist.

## Introduction

Shortly after SARS-CoV-2, the coronavirus that causes COVID-19, had emerged as a global threat to human health in January 2020, researchers had identified the host proteins required for viral entry into cells (*Hoffmann et al., 2020*; *Monteil et al., 2020*; *Wrapp et al., 2020*), repurposed drugs for treating COVID-19 patients (*Grein et al., 2020*; *Recovery Collaborative Group, 2020*), and initiated vaccine development (*Folegatti et al., 2020*; *Jackson et al., 2020*). A common feature of these advances was that they drew upon previous lines of research. A major question, however, is whether research into COVID-19 is pursuing all important host genes implicated in COVID-19.

To answer this question we used LitCOVID, a literature hub curated by the National Library of Medicine that tracks publications on COVID-19 (*Chen et al., 2020*). LitCOVID tags genes within the publicly accessible text of individual publications through PubTator (*Wei et al., 2019*), which first applies an ensemble of automated approaches to tag genes, and then allows for a revision of these tags through biocurators. We consider genes tagged within the title, abstract or results sections of individual publications, and use MEDLINE to exclude reviews and other non-research publications (see Methods). This yields 10,395 research publications featuring 3733 human protein-coding genes that have been tagged at least once. This enables us to ask whether the choices by scientists to investigate these genes can be understood in terms of current biological knowledge on COVID-19.

## Results

The most prominently tagged genes up to this point are: *Angiotensin-converting enzyme 2*, which serves as receptor for SARS-CoV-2 to enter cells (*Hoffmann et al., 2020*); *C-reactive protein*, a serum marker for inflammation (*Sproston and Ashworth, 2018*); and *Interleukin 6*, a mediator of systemic inflammatory responses (*Kang et al., 2019*). They account for 10.8%, 9.7%, and 4.5% of the total research on

human protein-coding genes within the COVID-19 literature, respectively (see Methods). Gene Ontology Enrichment analysis of the human protein-coding genes tagged in the COVID-19 literature finds them enriched for annotations on immune response (false-discovery rate $<10^{-66}$), inflammatory response (false-discovery rate $<10^{-65}$), and defense response to virus (false-discovery rate $<10^{-31}$) (*Supplementary file 1*). These two observations would thus suggest that the choice of host genes tagged in the COVID-19 literature is biologically grounded and in accord with current knowledge about respiratory viruses.

### Most host genes identified by genome-wide studies have not been pursued

Genome-wide datasets provide another window on SARS-CoV-2 infection. As genome-wide approaches circumvent research patterns that may have been pre-established within the scientific literature (*Haynes et al., 2018*; *Nelson et al., 2015*; *Stoeger et al., 2018*), they might identify additional genes implicated in COVID-19. RNA-sequencing (RNA-seq) was used recently to identify 1726 host genes that change the expression of their transcripts in the lungs of COVID-19 patients at an adjusted p-value<0.05 (*Blanco-Melo et al., 2020*). Affinity-purification mass spectrometry (Aff-MS) was used to identify 293 host proteins following the pulldown of exogenously expressed SARS-CoV-2 proteins (*Gordon et al., 2020*). Using genome-wide association studies (GWAS), the Host Genetics Initiative identified 52 genes through their association at a *P*-value of $10^{-5}$ or lower in one of three comparisons: COVID-19 vs lab or self-reported negative; hospitalized COVID-19 patients vs population; or very severe respiratory COVID-19 vs population (*COVID-19 Host Genetics Initiative, 2020a*; *Ellinghaus et al., 2020*). 15 genes were identified in two comparisons and one gene, *Leucine zipper transcription factor-like protein 1* (LZTFL1), was identified in all three comparisons (*Supplementary file 2*). Using a pooled CRISPR screen to affect SARS-CoV-2 induced cell death in African green monkey cells, Wei et al. identified 41 genes, which we mapped to their human homologs using BioMart (*Wei et al., 2020*; see Methods). 48 genes are identified in two of the four different genome-wide datasets (*Supplementary file 3*), but no gene is identified in more than two.

However, an analysis of the COVID-19 literature reveals that most (56%–71%) of the genes identified in these four datasets have not yet been tagged in the COVID-19 literature (*Figure 1A*). Thus, the genes identified by the four genome-wide datasets are 10–25% more likely to have been tagged than a randomly chosen gene because we also observe that 19% of all human protein-coding genes have been tagged at least once in the COVID-19 literature. Similarly, the fraction of tagged genes only increases by 0–7% if we include preprints (*Figure 1—figure supplement 1*). We conclude that many genes identified by genome-wide datasets on COVID-19 have not been investigated yet in more detail in the context of COVID-19.

At the same time, we observe that genes, which have been identified by multiple of the four distinct genome-wide datasets (*Figure 1B*), or multiple GWAS comparisons (*Figure 1C*), are more likely to have been tagged in the COVID-19 literature. This, reassuringly, demonstrates that research into COVID-19 host genes enriches for host genes identified by multiple different lines of support – particularly if there exists support from human genome-wide association studies.

Yet, overall, genes identified by multiple genome-wide datasets remain only a minority of all identified genes (2%), and many of them are still ignored in the COVID-19 literature (52%) (*Supplementary file 2*), suggesting that research into SARS-CoV-2 host genes might be missing important pieces of the puzzle.

### Tagged host genes follow pre-established research patterns

A possible explanation for the relative lack of interest in the additional genes implicated in SARS-CoV-2 infection by these genome-wide datasets is that research on COVID-19 is constrained by pre-established research patterns (*Chu and Evans, 2018*). Briefly, we know that knowledge on human genes is heavily skewed toward a subset of genes (*Gans et al., 2008*; *Gillis and Pavlidis, 2013*; *Hoffmann and Valencia, 2003*; *Oprea et al., 2018*; *Su and Hogenesch, 2007*) that were being investigated prior to the Human Genome Project (*Edwards et al., 2011*; *Grueneberg et al., 2008*; *Stoeger et al., 2018*). As a result, if assessing their importance

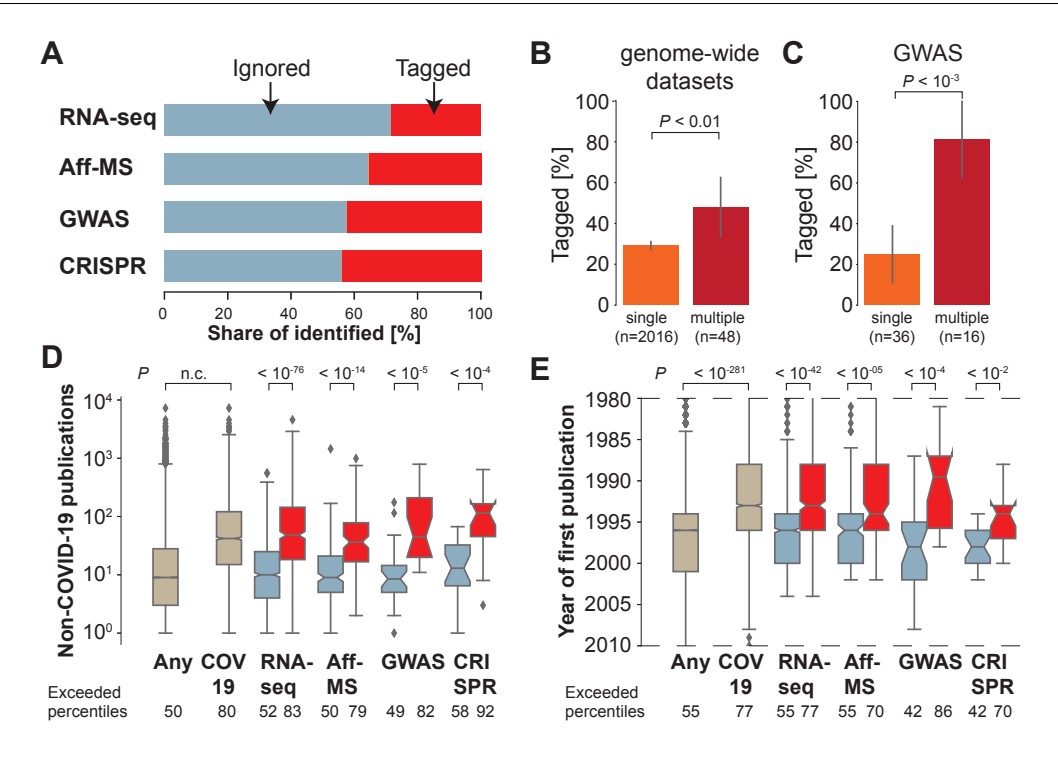

**Figure 1.** Most host genes implicated in COVID-19 identified by genome-wide approaches are not being investigated. (A) Share of identified genes, which are ignored (never tagged, blue) or tagged (at least once) within the COVID-19 literature. (B) Share of tagged genes identified by a single (orange) or multiple (maroon) genome-wide datasets. *P*-values are calculated via Fisher's exact test. n is the number of genes. (C) Share of tagged genes identified by a single (orange) or multiple (maroon) GWAS comparisons. *P*-values are calculated via Fisher's exact test. n is the number of genes. (D) Non-COVID-19 publications measured for any human protein-coding gene (ocher, any) and those occurring in the COVID-19 literature (ocher, COV19) and genes identified in A (colors as in A). Notches indicate 95% confidence interval of the median. *P*-values are calculated via *Mann-Whitney U* test. Exceeded percentiles indicates percentiles of all genes exceeded by the median gene of the genes in an individual boxplot. n.c. marks non-computable *P*-values that approximate 0. (E) As D, but for year of initial publication on the gene. Dashed lines indicate limit of visualized values. Some genes had their first publication before or afterwards.

The online version of this article includes the following figure supplement(s) for figure 1:

**Figure supplement 1.** Share of identified genes that are ignored or tagged.

---

through genetic loss-of-function intolerance or findings of GWAS (*Haynes et al., 2018*; *Stoeger et al., 2018*), the most frequently investigated protein-coding genes tend to be only marginally more important to human physiology and disease than a random selection of genes.

To test the hypothesis that COVID-19 research is constrained by patterns similar to those seen in non-COVID-19 research, we take advantage of the ability of gene2pubmed (a service provided by the National Center for Biotechnology Information) to link human protein-coding genes to individual publications, and compare 465,770 non-COVID-19 papers published until December 2015 with 10,395 COVID-19 research publications indexed by LitCOVID

until October 16th, 2020. For the non-COVID-19 research we exclude publications that contain any viral gene (irrespective of whether the virus in question is a coronavirus) and publications tagging 100 or more genes.

We find that genes that are tagged in the COVID-19 literature are also frequently investigated in the non-COVID-19 literature. To assess how frequently individual genes have been investigated in the non-COVID-19 literature relative to other genes, we rank all genes according to the number of publications in the non-COVID-19 literature. The median rank of genes tagged in the COVID-19 literature exceeds the rank of 80% of human protein-coding genes (*Figure 1D*). This demonstrates that the majority

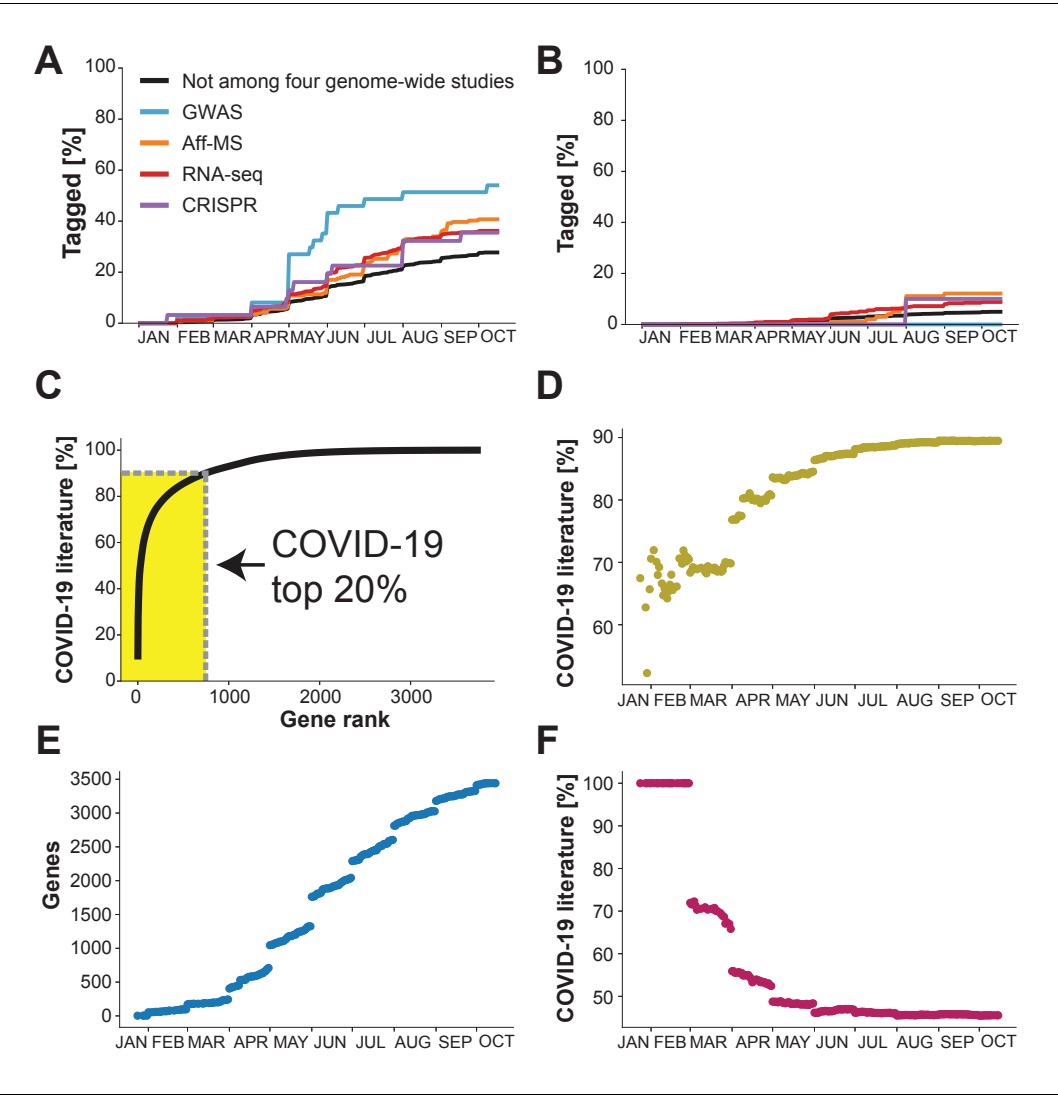

**Figure 2.** What the future holds? Percentage of genes with indicated levels of support by the four genome-wide studies which have been tagged at least once in the COVID-19 literature. (**A**) Analysis restricted to the 50% of genes with highest number of publications in non-COVID-19 literature. (**B**) Analysis restricted to the 50% of genes with the lowest number of publications in the non-COVID-19 literature. (**C**) Cumulative share of literature on human protein-coding genes tagged in the COVID-19 literature. Top 20% indicates the 20% of genes that occur the most in the non-COVID-19 literature. Gene rank refers to the order of human protein-coding genes. The gene with the most publication equivalents would be have rank 1. Yellow area indicates share of literature accounted for by the top 20% genes. (**D**) Share of COVID-19 literature accounted for by the 20% of genes that had occurred the most in the COVID-19 literature by a given date. (**E**) Number of distinct human protein coding genes that have been tagged in the literature by a given date. (**F**) Share of COVID-19 literature accounted for by first 100 genes to be tagged in the COVID-19 literature by a given date.

The online version of this article includes the following figure supplement(s) for figure 2:

**Figure supplement 1.** Temporal trends in the diversity of COVID-19 research.

of protein-coding human genes tagged in the COVID-19 literature was already heavily investigated in the context of research unrelated to COVID-19.

Next we return to our earlier observation on the majority of the implicated host genes reported by the four different genome-wide datasets being ignored within the COVID-19 literature. As anticipated, we observe that for each of the four distinct datasets investigated, ignored genes also occur less in this non-COVID-19 literature (*Figure 1D*). When we

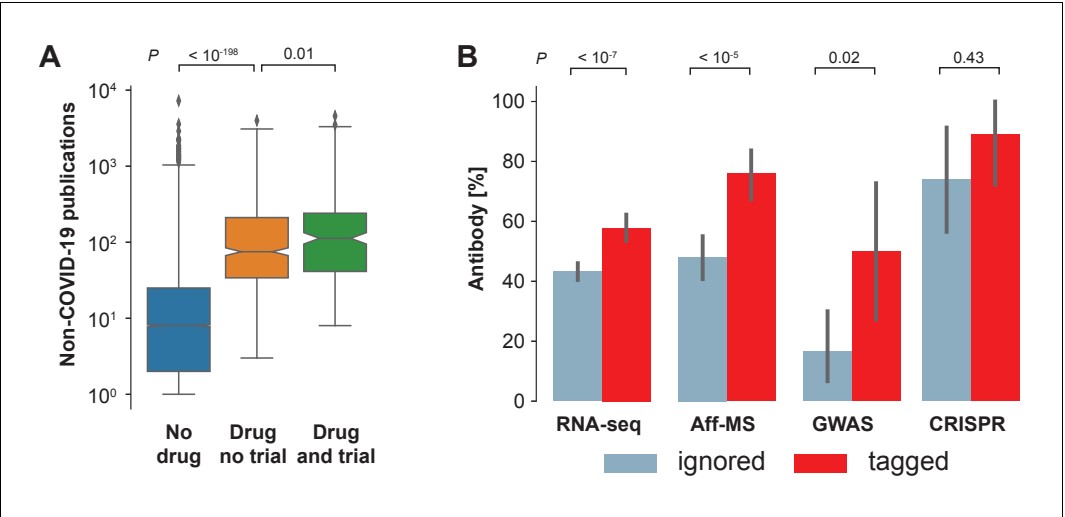

**Figure 3.** Availability of reagents. (A) Drugs studied in COVID-19 related clinical trials are frequently studied within the non-COVID-19 literature. We compare non-COVID-19 publications measured for human protein-coding genes that are not listed as pharmaceutical targets in DrugBank (ocher, No drug), against those that are listed as pharmaceutical targets but have not occurred as an intervention in a clinical trial on COVID-19 (orange, Drug no trial), and against those that are listed as pharmaceutical targets and have occurred as an intervention in a clinical trial on COVID-19 (green, Drug and trial). Notches indicate 95% confidence interval of the median. *P*-values are calculated via *Mann-Whitney U* test. (B) Fraction of genes with reported usage of an antibody to detect the encoded protein as a prey in BioGRID. Bars are genes identified by the four different genome-wide studies that have either been tagged in the COVID-19 literature (red) or ignored (blue). Error bars indicate 95% confidence interval. *P*-values are calculated via Fisher's exact test.

compare the number of publications on implicated but ignored host genes to the number of publications on any protein-coding gene encoded in the human genome, this difference is modest, and only reaches statistical significance for RNA-seq (RNA-Seq: $p<10^{-2}$; Interactomics: p=0.20; GWAS: p=0.93; CRISPR: p=0.31), where ignored genes had occurred slightly more in the non-COVID-19 literature (median percentile: 52). In contrast, implicated and tagged host genes have occurred significantly more frequently in the non-COVID-19 literature (RNA-Seq: $p<10^{-98}$; Interactomics: $p<10^{-18}$; GWAS: $p<10^{-6}$; CRISPR: $p<10^{-6}$). We conclude that implicated host genes that are ignored in the COVID-19 literature have in the past been studied as much as randomly chosen human protein-coding genes, whereas implicated host genes that are tagged in the COVID-19 literature have in the past already been investigated much more frequently than randomly chosen human protein-coding genes.

Before the COVID-19 pandemic it had been shown that the literature is skewed toward a subset of genes that were being investigated prior to the productive phase of the Human Genome Project. These features include the fraction of organs with detectable transcript expression, the length of the genes, the hydrophobicity of the coded proteins, their loss-of-function insensitivity, and studies on orthologous genes in model organisms (*Stoeger et al., 2018*). We decided to explore if the genes tagged in the COVID-19 literature had been studied before the pandemic, and found that they had occurred earlier (*Figure 1E*), with many also first being studied before the productive phase of the Human Genome Project (*NHGRI, 2003*). Similarly, the host genes identified by the four genome-wide datasets that are ignored in the COVID-19 literature first appeared in the non-COVID-19 literature after the host genes that are tagged in the COVID-19 literature (*Figure 1E*).

### Trends over time

The COVID-19 pandemic has ravaged for less than a year, which is a short period of time compared to most research projects. Thus, we might not yet be observing research addressing poorly-studied implicated host genes because not sufficient time has passed for research to catch up to the new information.

To anticipate the near future, we follow the occurrence of genes in the COVID-19 literature

over time. Based on our insight that ignored host genes have not been studied more than other genes in the non-COVID-19 literature (*Figure 1D*), we separate genes into two classes: genes that are among the 50% top-studied human protein-coding genes in the non-COVID-19 literature, and genes that are among the 50% least-studied human protein-coding genes. The second class holds 35% of the genes identified by RNA-seq, 33% of the genes identified by Aff-MS, 29% of the genes identified by GWAS, and 24% of the genes identified by CRISPR. If research is catching up to the new knowledge, we would expect to see the fraction of the COVID-19 literature addressing the 50% least-studied human protein-coding genes to increase over time.

When focusing on the genes that are among the 50% top-studied human protein-coding genes, we observe their occurrence in the COVID-19 literature to increase steadily. Extrapolating from the observed trends, we anticipate that it will take around one year till nearly all genes of this class will have been tagged at least once within the COVID-19 literature (*Figure 2A*). When focusing on the genes that are among the 50% least-studied human protein-coding genes, we too observe their occurrence in the COVID-19 literature to increase steadily over time (*Figure 2B*). As for each of the four genome-wide datasets the increase is, however, slower than for the 50% most studied protein-coding genes, we project that multiple years could pass until each gene of the 50% least-studied human protein-coding genes will have been tagged at least once within the COVID-19 literature.

Pursuing this observation further, we turn to the entire COVID-19 literature. Notably, 83% of all human protein-coding genes tagged in the COVID-19 literature have not been identified by any of the four genome-wide datasets. Further, the different genome-wide datasets together only account for 26% of the COVID-19 literature (RNA-seq: 11.7%, Aff-MS: 2.4%, GWAS: 0.5%, CRISPR: 11.1%) (see Methods).

We ask whether the COVID-19 literature might become dominated by a few genes that are tagged more commonly than other genes that are also tagged in the COVID-19 literature. If we consider the current literature, we do indeed observe support for our hypothesis that the COVID-19 literature is becoming dominated by a few genes as currently the 20% top-tagged human protein-coding genes (747 of 3,733) in the COVID-19 literature account for 90% of the literature (*Figure 2C*). This share exceeds the

80% anticipated for scientific processes subjected to anthropogenic biases (*Jia et al., 2019*). We conclude that a surprisingly small fraction of genes dominates the COVID-19 literature.

Finally, we inspect whether the extent to which the COVID-19 literature tags each tagged gene is becoming more or less expansive over time. We observe that the COVID-19 has become less expansive, whether we quantify expansiveness through the share of the literature that is accounted for by the 20% top-tagged genes or the Gini coefficient over the share of the COVID-19 literature attributable to individual genes (*Gini, 1912*; *Figure 2D*, *Figure 2—figure supplement 1A,B*). However, if assessing expansiveness by the total number of genes that have been tagged at least once, then this literature did become more expansive after the first months (*Figure 2E*).

Interestingly, we also observe that the share of the COVID-19 literature, which is accounted for by the 100 genes that were tagged first within the COVID-19 literature has been decreasing (*Figure 2F*, *Figure 2—figure supplement 1C*) – though stabilizing at an astonishingly high share of roughly 45% since June (*Figure 2F*, *Figure 2—figure supplement 1C*). We conclude that, overall, the literature on COVID-19 became less expansive during the first months of the pandemic and has since stayed focused on a restricted subset of genes.

One possible reason for why some genes are tagged more than others in the COVID-19 literature could be that compared to other genes they are more important in the context of COVID-19. To probe this hypothesis, we consider groups of genes and the four different genome-wide datasets. When contrasting the 100 initially tagged genes against the other genes tagged in the COVID-19 literature, we reassuringly find that the 100 initially tagged genes are 29% more likely to have been identified by one of these four datasets (*Supplementary file 4*). However, they are on average tagged 2993% more (*Supplementary file 4*). If we contrast the 20% top-tagged genes against the other tagged genes, we find them to be 3% less likely to have been identified by one of the four genome-wide datasets, while they on average are tagged 3512% more (*Supplementary file 4*). Cumulatively, this suggests that the present focus of the COVID-19 literature on a restricted subset of genes cannot be explained by those genes having been identified by genome-wide datasets

reporting on transcriptomic changes, protein interactions, genetic associations and loss-of-function perturbations.

### Study limitations

Our study has several important limitations. First, we cannot say whether a gene tagged in the COVID-19 literature is truly investigated for its potential role in COVID-19. Second, we cannot yet assess how important individual genes are in COVID-19. Third, and despite our projections, it remains formally unclear, whether the findings reported in this manuscript will hold in the upcoming months as more genome-wide datasets will become available and researchers will have had sufficient time for follow-up studies. For instance, there might already be research initiatives specifically targeted toward the ignored COVID-19 host genes.

Nonetheless, in the past, genome-wide experiments have rarely guided subsequent studies in the non-COVID-19 literature (*Haynes et al., 2018*; *Stoeger et al., 2018*). Thus, there is a significant risk that the COVID-19 literature will continue to ignore host genes that have not already been extensively studied independently of COVID-19.

## Discussion

Our study reiterates prior observations that research into human protein-coding genes is disproportionately skewed towards a comparably small set of genes (*Haynes et al., 2018*; *Nelson et al., 2015*; *Stoeger et al., 2018*). Likewise, our current analysis on COVID-19 already allows us to conclude that genes that are identified by genome-wide datasets, and hence are likely to have biological significance in the context of COVID-19, have hitherto remained ignored if they had not already been investigated more than other genes prior COVID-19.

We realize that there is an exploration-exploitation trade-off at play and that focusing research on genes that have already been heavily investigated yields significant advantages to investigators: applicability of existing research tools, the ability to place findings in a broader context, and the identification of drugs and other reagents that could be repurposed. Supporting a focus on exploitation, we find that: interventions in clinical trials on COVID-19 are biased toward pharmaceutical targets that occurred frequently in the non-COVID-19 literature (*Figure 3A*); and that antibodies – a class of reagent that cannot be produced for arbitrary

genes within a few days – are less available for those genes identified by RNA-seq or Aff-MS or GWAS which have been ignored in the COVID-19 literature (*Figure 3B*).

Further, additional factors might affect the exploration of studies on ignored host genes. First, the number of laboratories working on ignored genes was quite small prior COVID-19 (*Supplementary file 5*), and plausibly only a small fraction of the laboratories studied host responses toward respiratory viruses. Second, the risk of being outcompeted by other laboratories might discourage individual laboratories from pursuing publicly acknowledged research targets (*Bergstrom et al., 2016*). Third, scientists rarely switch topics (*Zeng et al., 2019*). Likewise, laboratories already working on COVID-19 might have little incentive to move toward distinct host genes as it is possible to contribute to the COVID-19 literature irrespectively of whether the genes had been identified by genome-wide datasets (*Figure 2A,B* and *Supplementary file 4*). Moreover, it might be beneficial overall if research into COVID-19 is mainly driven by researchers with a background on pathogens (*Kwon, 2020*). Lastly, concerns have been expressed about the possibility that fraudulent gene knockdown studies that target under-studied human genes may be corrupting the literature and impeding research into biomarkers (*Byrne et al., 2019*).

We believe that a more complete understanding of host biology could open novel directions for interventions against SARS-CoV-2 and other viruses. However, the challenge remains of how to promote research on ignored host genes. For example, we cannot speculate whether researchers that turn their attention toward ignored genes in the context of COVID-19, will face a similar disadvantage to their career as did those that studied less studied genes prior to COVID-19 (*Stoeger et al., 2018*).

In the hopes of prompting greater investigation into implicated host genes, we list genes occurring in multiple of the four datasets described earlier in the supplemental material of this manuscript (*Supplementary file 3*). Most of the genes identified by multiple datasets appear multiple times because of the large volume of genes identified by the RNA-seq dataset. For this reason, we highlight four genes that were identified by multiple smaller datasets: (1) *Mitochondrial import inner membrane translocase subunit Timm10* (TIMM10) has been identified through Aff-MS and CRISPR, and (2) *FYVE And Coiled-Coil Domain Autophagy Adaptor 1*

(FYCO1) and (3) *Procollagen-Lysine,2-Oxoglutarate 5-Dioxygenase 2* (PLOD2) and (4) Ras GTPase-activating protein-binding protein 2 (G3BP2) have been identified through Aff-MS and GWAS. Of these four genes only G3BP2 has been tagged in the COVID-19 literature; and TIMM10 and FYCO1 have both occurred in nine publications in the non-COVID-19 literature, matching the expectation for a randomly selected gene. Of additional interest in the context of COVID-19, FYCO1 is associated with the levels of the monocyte chemoattractant protein-1 (*Ahola-Olli et al., 2017*; *Buniello et al., 2019*), which contributes to COVID-19 through hyperinflammation (*Mehta et al., 2020*).

## Methods

### COVID-19 literature

We downloaded LitCOVID from https://ftp.ncbi.nlm.nih.gov/pub/lu/LitCovid/ on 2020-10-16 and parsed the contained json file for the presence of concepts annotated as genes. For studies annotated with proteins, we used their PubMed identifiers, to query MEDLINE on 2020-10-16 via their efetch API. Subsequently we parsed the MEDLINE entries via pubmed_parser 2.2 (https://github.com/titipata/pubmed_parser; *Achakulvisut et al., 2020*). We then excluded publications carrying at least one of the following publication types: Review, Comment, Editorial, Meta-Analysis, Systematic Review, News, Published Erratum, Historical Article, Interview, Retracted Publication, Retraction of Publication, Webcast, Expression of Concern or Portrait. Further we excluded publications whose abstract contain either the phrase 'this review' or the phrase 'this perspective'. We considered genes tagged within LitCOVID in the annotated TITLE, INTRO, ABSTRACT or RESULTS sections.

### Research intensity within COVID-19 literature

We measured the research intensity directed toward individual implicated host genes in units of publication equivalents. Each gene tagged within a publication accrues the publication equivalent of number of tags to the gene in that publication divided by total number of tags to any gene in that publication. For example, if a study tags two different genes, and the first gene is tagged three times, whereas the second gene is only tagged once, the first gene would accrue 0.75 publication equivalents, and the second gene would accrue 0.25 publication

equivalents. We expressed the share of literature covered by an individual human protein-coding gene as the sum of its publication equivalents over the sum of all publication equivalents of human protein-coding genes. We excluded the studies of *Blanco-Melo et al., 2020* and *Gordon et al., 2020* which report the RNA-seq and Aff-MS datasets, respectively.

### Gene ontology enrichment analysis

We used the Database for Annotation, Visualization and Integrated Discovery, version 6.8 (*Huang et al., 2009*).

### Data processing and filtering

For CRISPR we considered the African green monkey genes reported in Figure 1D of *Jin Wei et al., 2020*. To map African green monkey to human genes, we used BioMart's (*Haider et al., 2009*) April 2020 release. We used the genetic polymorphisms reported in the 2020-09-30 release of the Host Genetics Initiative (*COVID-19 Host Genetics Initiative, 2020b*) and mapped them to human genes through the Ensembl Variant Effect Predictor (*McLaren et al., 2016*), using the Ensembl release 101. For RNA-seq we only considered comparisons flagged with 'ok' by the authors (*Blanco-Melo et al., 2020*). For Aff-MS we used the data as provided by BioGRID (*Chatr-Aryamontri et al., 2017*), version 3.5.186 (https://downloads.thebiogrid.org/BioGRID).

We obtained the list of human protein-coding genes from https://ftp.ncbi.nlm.nih.gov/gene/DATA/gene_info.gz in June 2020.

### Occurrence of genes in preprints

We obtained manuscripts abstracts from dimension.ai's collection of COVID-19 related publications, release 34 (https://dimensions.figshare.com/articles/dataset/Dimensions_COVID-19_publications_datasets_and_clinical_trials/11961063/34; *dimension.ai, 2020*), and subsequently select manuscripts listing medRxiv or bioRxiv or arXiv as their source. Next, we matched each word against the gene symbols as downloaded from https://ftp.ncbi.nlm.nih.gov/gene/DATA/gene_info.gz in June 2020.

We excluded the following gene symbols as within the abstracts they would match abbreviations that did not refer to genes: AFM, AIR, AN, APC, APP, AR, ARC, ATM, BCR, BED, BID, CCNC, CFD, CHM, COPD, COPE, CP, CPE, CS, DBI, DCT, ENG, GAN, GC, HP, HPA, HPD, HPO,

HR, IDS, IMPACT, IV, KIT, MCC, MET, MICE, MMD, MS, MS2, NHS, NM, NPS, NSF, NTS, PIP, POLL, REST, SEA, SET, SHE, SI, SPR, STS, TAT, TRAP, WAS.

### Non-COVID-19 literature

We downloaded gene2pubmed from https://ftp.ncbi.nlm.nih.gov/gene/DATA/gene2pubmed.gz in early 2017. MEDLINE, containing publication dates and publication types was downloaded from https://www.nlm.nih.gov/databases/download/pubmed_medline.html, and maintained in a local copy of their database in early 2017. We restricted the analysis to research publications published prior 2016.

### Temporal profiles

We obtained publication dates from dimension.ai's collection of COVID-19 publications, release 34 (https://dimensions.figshare.com/articles/dataset/Dimensions_COVID-19_publications_datasets_and_clinical_trials/11961063/34). We excluded publications dating to January 1st, 2020 – the day linked to the most publications. Manual inspection revealed that the date of January 1st 2020 was assigned to publications lacking a concrete 2020 publication date.

### Occurrence within clinical trials

We obtained interventions within clinical trials from dimension.ai's collection of COVID-19 related clinical trials, release 34 (https://dimensions.figshare.com/articles/dataset/Dimensions_COVID-19_publications_datasets_and_clinical_trials/11961063/34; *dimension.ai, 2020*). We performed a case-insensitive match against drug names and drug synonyms contained within DrugBank, version 5.1.5 (https://www.drugbank.ca). Next we used DrugBank's mapping between drugs and the targets of their pharmaceutical action and used the accompanying gene symbol to identify genes.

### Identification of research laboratories

We used disambiguated authorship identifiers from Web of Science and considered the last author of each publication as the laboratory.

### Acknowledgements

We thank all members of the Amaral lab for feedback, particularly Jennifer Liu, Kedi Cao, Meagan Bechel, Reese Richardson and Sarah Ben Maamar. We thank Richard Wunderink for feedback and suggesting the analysis of clinical trials. We also thank Rick Morimoto for feedback.

**Thomas Stoeger** is in the Successful Clinical Response in Pneumonia Therapy (SCRIPT) Systems Biology Center, the Department of Chemical and Biological Engineering and the Northwestern Institute on Complex Systems (NICO), Northwestern University, Evanston, United States, and the Center for Genetic Medicine, Northwestern University School of Medicine, Chicago, United States

thomas.stoeger@northwestern.edu

https://orcid.org/0000-0002-5540-4278

**Luís A Nunes Amaral** is in the Successful Clinical Response in Pneumonia Therapy (SCRIPT) Systems Biology Center, the Northwestern Institute on Complex Systems (NICO), the Department of Molecular Biosciences, and the Department of Physics and Astronomy, Northwestern University, Evanston, United States, and the Department of Medicine, Northwestern University School of Medicine, Chicago, United States

amaral@northwestern.edu

https://orcid.org/0000-0002-3762-789X

*Author contributions:* Thomas Stoeger, Conceptualization, Resources, Data curation, Software, Formal analysis, Funding acquisition, Investigation, Visualization, Methodology, Writing - original draft, Project administration, Writing - review and editing; Luís A Nunes Amaral, Conceptualization, Resources, Formal analysis, Supervision, Funding acquisition, Investigation, Visualization, Methodology, Writing - original draft, Project administration, Writing - review and editing

*Competing interests:* The authors declare that no competing interests exist.

#### Funding

| Funder | Grant reference number | Author |
|---|---|---|
| National Institute of Allergy and Infectious Diseases | U19AI135964 | Luís A Nunes Amaral |
| National Science Foundation | 1956338 | Luís A Nunes Amaral |
| Simons Foundation | DMS-1764421 | Luís A Nunes Amaral |
| Air Force Office of Scientific Research | FA9550-19-1-0354 | Luís A Nunes Amaral |
| National Institute on Aging | K99AG068544 | Thomas Stoeger |

The funders had no role in study design, data collection and interpretation, or the decision to submit the work for publication.

### Decision letter and Author response

Decision letter https://doi.org/10.7554/eLife.61981.sa1
Author response https://doi.org/10.7554/eLife.61981.sa2

## Additional files

### Supplementary files

• Source code 1. Source code for curation and analysis of datasets.

• Supplementary file 1. Gene Ontology enrichment analysis for human protein-coding genes tagged in the COVID-19 literature.

• Supplementary file 2. Identification of genes through multiple GWAS comparisons.

• Supplementary file 3. Implicated host genes identified by multiple genome-wide studies.

• Supplementary file 4. Extent of tags in COVID-19 literature compared to rate identification in genome-wide datasets. Per-gene average share of COVID-19 literature and per-gene average identification rate in genome-wide datasets (one if identified, 0 if not identified). Shown are the ratios of this share and the rates in individual groups (100 first tagged genes, and 20% top-tagged genes) over the share and the rates of the other genes that have been tagged in the COVID-19 literature.

• Supplementary file 5. Number of laboratories working on individual genes, identified within one of the four genome-wide datasets, between 2006 and 2015.

• Transparent reporting form

### Data availability

No data was generated for this study. Data underlying this study can be downloaded from the sources indicated in the methods section and used under the respective licenses. The data can be preprocessed with the source code accompanying this manuscript, and - for literature until 2015 - the public source code provided in a former publication of ours, https://github.com/tstoeger/plos_biology_2018_ignored_genes.

The following datasets were generated:

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
