## [Decision Letter]

Thank you for submitting your article "Meta-Research: COVID-19 research risks ignoring important host genes due to pre-established gene-specific biases" to eLife for consideration as a Feature Article. Your article has been reviewed by three peer reviewers, and the evaluation has been overseen by the eLife Features Editor. The following individuals involved in review of your submission have agreed to reveal their identity: Valentin Danchev (Reviewer #1); Hong Zheng (Reviewer #2); Steve Brown (Reviewer #3).

The reviewers and editors have discussed the reviews and we have drafted this decision letter to help you prepare a revised submission.

Summary:

There is now a very considerable body of evidence, much if it provided and substantiated through the work of Stoeger and Nunes, that the majority of published gene to phenotype studies continue to focus on genes that are well-annotated or for which knowledge of biological function and pathological consequences of mutations already exist. As a result, much of the human genome remains unexplored and considered "dark". This is not just a major problem for the dynamic development of the biological and biomedical sciences and the provision of novel insights into biological and disease mechanisms. It has the potential to impact significantly on the assessment of mechanisms and interventions in scenarios such as pandemics arising from novel pathogens which potentially elicit and require investigation and understanding of novel host-pathogen interactions involving unknown genetic pathways. Stoeger and Nunes now show, based on an analysis of the COVID-19 literature up to 30 July 2020, that COVID-19 studies display similar biases and critically ignore recent genome-wide datasets. However, a number of points need to be addressed to make the article suitable for publication.

Essential revisions:

1) Please update the analysis to include COVID-19 papers published until the end of August 2020 (or later).

2) The comparison of published literature to large datasets from GWAS experiments is interesting but a comparison of the GWAS datasets themselves would add some interesting detail as well. What is the percentage of overlapping genes across all four GWAS? Is the literature more likely to select for study those genes that appear in many or all four GWAS? If it is not possible to answer these questions, please discuss this issue in the text.

3) Published papers as of July 30 are considered, many of them likely written in May or early June, i.e., a couple of months after the WHO characterised COVID-19 as pandemic. This appears to be bias away from the null. Realistically, it would take some time for the literature to diversify. Preprints could have provided a more nuanced assessment but admittedly, extracting gene keywords from preprints would be computationally hard.

A longitudinal analysis (e.g., two week slices) could be informative – is it the case that earlier in time labs started from diverse sets of genes and then converged; or, the other way around, they started from a few candidates, and then diverged. Or, there was no significant change over time, with incremental increase of the number of genes as the number of papers increased.

If it is not possible to conduct such a longitudinal analysis, please discuss this issue in the text.

4) The framing of the problem in terms of bias might be too strong – it's just too early to tell. An alternative, and less charged ways of framing the problem would be in terms of the exploration-exploitation trade-off.

Essentially, the manuscript tells us that labs have continued to study the genes they studied before COVID-19. The manuscript views this as a bias. But is it? A more charitable interpretation would be that labs continued to study what they know rather than jumping on the new fashionable possibility. A major concern since the pandemic has been the tendency for many to assert expertise in new areas relevant to Covid-19, and it seems that genomic labs are less involved in this. Could that path-dependency bias be seen a good thing in the broader context?

5) It would be helpful to have a discussion on how realistic it is for a lab studying a set of genes to start studying another set of genes at short notice. Studies of country's manufacturing and other fields suggest that big "jumps" in the space of possibilities are limited, meaning that one could jump from B to D but hardly to X, Y, or Z. The manuscript should clarify how possible it is in the context of those gene experiments for a lab to study one set of genes and then "jump" to another set. In other words, if a lab would like to avoid the bias the manuscript outlines, how easy would be to study those understudied genes in terms of expertise, lab equipment, cell lines?

6) Regarding the "Pareto principle" that 80% of research falls onto only 20% of the genes, have the authors considered the possibility that the "Pareto principle" might also apply to the importance of genes? The coding sequence is only 2% of the human genome, which is more extreme than the 80/20 ratio. Within the protein coding genes, the transcriptional factors or the hub of networks may just be functionally more important than other genes. Thus, is it possible that the observed bias in research reflects the innate features of human genes?

7) Another factor that might contribute to the bias is how easy it is to study the gene, especially in early days when the first step is to clone a gene, and the longer the gene is, the harder the process is. Could the authors look into whether the bias is partly introduced by the length of the genes?

If it is not possible to conduct such an analysis, please discuss this possibility in the text.

---

## [Author Response]

Essential revisions:1) Please update the analysis to include COVID-19 papers published until the end of August 2020 (or later).

We updated the manuscript for studies identified until October 16th.

2) The comparison of published literature to large datasets from GWAS experiments is interesting but a comparison of the GWAS datasets themselves would add some interesting detail as well. What is the percentage of overlapping genes across all four GWAS? Is the literature more likely to select for study those genes that appear in many or all four GWAS? If it is not possible to answer these questions, please discuss this issue in the text.

While at the time at the initial submission there was no overlap, the newest release of the GWAS catalog finds 15 genes in two comparisons, and one gene, LZTFL1, in all three comparisons. We now report so in the text and the new Supplementary File 2.

We can now also confirm the reviewer’s hypothesis that genes occurring in multiple GWAS comparisons are more likely to have been mentioned in the COVID-19 literature (Figure 1C). Motivated by this finding we further inspected whether genes identified by multiple genome-wide datasets too are more likely to have been mentioned in the COVID-19 literature – which in turn they are (Figure 1B). However, these genes still only account for a minority of the genes in the COVID-19 literature, and several genes that have been identified by multiple distinct GWAS comparisons or genome-wide datasets remain to be mentioned in the COVID-19 literature.

3) Published papers as of July 30 are considered, many of them likely written in May or early June, i.e., a couple of months after the WHO characterised COVID-19 as pandemic. This appears to be bias away from the null. Realistically, it would take some time for the literature to diversify.

To emphasize this important point, we added a new section: “What the future holds?”

Preprints could have provided a more nuanced assessment but admittedly, extracting gene keywords from preprints would be computationally hard.

We now additionally analyze preprints. Our findings are not altered.

A longitudinal analysis (e.g., two week slices) could be informative – is it the case that earlier in time labs started from diverse sets of genes and then converged; or, the other way around, they started from a few candidates, and then diverged. Or, there was no significant change over time, with incremental increase of the number of genes as the number of papers increased.

We now include a temporal analysis of distinct measures of diversity (Gini coefficient, number of genes, share of literature accounted for by 20% most mentioned genes, share of literature accounted for by the 100 genes first mentioned in the COVID-19 literature) and inspect them twice (Figure 2D-F, Figure 2—figure supplement 1). First, cumulatively until individual days. Second, separately for each month (instead of two-week slices as some applied measures of diversity would look wavy in the latter due to the higher number of publications published at the beginning of each month). Briefly, we find that since June the literature has reached what appears to be a stationary state.

To further strengthen our analysis, we now also include a longitudinal analysis of the number of genes that had been identified by the four genome-wide datasets (Figure 2A, B). It shows that genes that had not been commonly studied in the non-COVID-19 literature are introduced into the COVID-19 literature at a slower pace than those genes that had been commonly studied in the non-COVID-19 literature.

If it is not possible to conduct such a longitudinal analysis, please discuss this issue in the text.

We thank the reviewer for having suggested a longitudinal analysis and believe that it has strengthened our manuscript.

4) The framing of the problem in terms of bias might be too strong – it's just too early to tell. An alternative, and less charged ways of framing the problem would be in terms of the exploration-exploitation trade-off.

We reframed the manuscript. First, we avoided the terminology “bias”. Second, we now explicitly discuss the trade-off between exploration and exploitation and extended the analysis (Figure 3B) to demonstrate that antibodies – one class of gene specific reagents that cannot be manufactured within a few days – are more likely to exist for identified genes that were also mentioned in the COVID-19 literature (a finding that is significant for RNA-seq, Aff-MS and GWAS but not CRISPR).

Essentially, the manuscript tells us that labs have continued to study the genes they studied before COVID-19. The manuscript views this as a bias. But is it? A more charitable interpretation would be that labs continued to study what they know rather than jumping on the new fashionable possibility.

We now extended the Discussion section to list reasons why laboratories might not jump to new possibilities.

A major concern since the pandemic has been the tendency for many to assert expertise in new areas relevant to Covid-19, and it seems that genomic labs are less involved in this. Could that path-dependency bias be seen a good thing in the broader context?

We now mention this possibility. We further explicitly mention a commentary of another group which points out that the influx of scientists from unrelated fields could risk compromising the integrity of the literature and make it more difficult to identify valuable publications.

5) It would be helpful to have a discussion on how realistic it is for a lab studying a set of genes to start studying another set of genes at short notice. Studies of country's manufacturing and other fields suggest that big "jumps" in the space of possibilities are limited, meaning that one could jump from B to D but hardly to X, Y, or Z. The manuscript should clarify how possible it is in the context of those gene experiments for a lab to study one set of genes and then "jump" to another set. In other words, if a lab would like to avoid the bias the manuscript outlines, how easy would be to study those understudied genes in terms of expertise, lab equipment, cell lines?

We now extended the discussion to indicate that we anticipate switching to another set of genes is unlikely due to the risk of being outcompeted by other laboratories, the low availability of reagents for ignored genes, and the low baseline probabilities of scientists to switch topics. While these arguments are based on preceding studies of science, we anticipate them to also hold true for COVID-19.

Moreover, we now provide the number of laboratories working on ignored genes pre-COVID-19 to demonstrate that without big jumps toward these genes, there would be a limited potential of laboratories (likewise, making a jump toward COVID-19 might be a big jump for many of these laboratories) (Supplementary file 5).

6) Regarding the "Pareto principle" that 80% of research falls onto only 20% of the genes, have the authors considered the possibility that the "Pareto principle" might also apply to the importance of genes? The coding sequence is only 2% of the human genome, which is more extreme than the 80/20 ratio. Within the protein coding genes, the transcriptional factors or the hub of networks may just be functionally more important than other genes. Thus, is it possible that the observed bias in research reflects the innate features of human genes?

We now directly acknowledge this limitation within the section on study limitations. We believe that currently it is not possible to quantify the extent of the importance of individual genes toward COVID-19. Indeed, there may be multiple dimensions of importance relating to fraction of cells affected, severity of disease outcomes, number of patients affected, and the relation between these dimensions and the molecular findings is still unclear.

At the same time, we now introduce a novel analysis, Supplementary file 4, which demonstrates that – as a group – the genes that have been studied the most or the first account for a disproportional ~30 times larger share of the COVID-19 literature than one would anticipate based on the propensity of them having been identified by the four genome-wide datasets.

Further we extended the discussion section to add a statement that the identification of genes in genome-wide datasets suggests that they have some importance.

7) Another factor that might contribute to the bias is how easy it is to study the gene, especially in early days when the first step is to clone a gene, and the longer the gene is, the harder the process is. Could the authors look into whether the bias is partly introduced by the length of the genes?If it is not possible to conduct such an analysis, please discuss this possibility in the text.

We now add an analysis that demonstrates that genes studied in the COVID-19 literature had already been studied prior the productive phase of the Human Genome Project (Figure 1E). Given our earlier findings (Stoeger et al., 2018) this could be mainly interpreted as these genes having been easier to study within the early days.

For gene length as a specific hypothesis, there is additional complexity, and we would thus prefer not to include it. First, length might be less important than other factors facilitating experimentation (such as protein abundance in HeLa cells, or orthologs in model organisms, etc). Regretfully we noted that a statistical approach, which considers multiple alternate possibilities (and we used in Stoeger et al., 2018) is not yet applicable to the COVID-19 literature as it has no predictive power. We believe that this mainly stems from the comparably low number of COVID-19 publications compared to the number of publications in the non-COVID-19 literature. Second – and we did not report so explicitly before – length has a non-linear relationship to publications within the non-COVID-19 literature. Though the earliest discovered genes tend to be short, the longest genes too are studied more than genes with an “average” length (but still studied less than the shortest genes).

Below we add an analysis of gene lengths, analogously to Figures 1D,E from the main manuscript.
